# REASONING SUPERVISION FOR VISION TRANSFORMERS IN HUMAN ACTIVITY RECOGNITION

## ABSTRACT

Recent reasoning methods have been explored to improve model transparency and trust, particularly in video understanding, where actions are defined by temporal order, object interactions, and state transitions. However, most approaches remain post-hoc, offering limited opportunity to influence a model's internal reasoning process or improve its accuracy. In this work, we move beyond post-hoc explanation and introduce a Reasoning Supervision training pipeline that directly enhances model performance. This setting presents unique challenges: how to generate training-time reasoning guidance, what form this guidance should take, and how to inject it effectively into the model. Our framework addresses these challenges by leveraging large language models (LLMs) as proxy annotators to generate high-quality spatial supervision. We introduce two complementary loss functions to inject this guidance into the model: a spatial alignment loss that aligns attention with LLM-derived spatial reasoning guidance and a temporal reasoning loss that encourages coherent, human-like temporal dependencies across frames. Applied to Vision Transformer architectures, Reasoning Supervision consistently improves performance, establishing a simple yet effective paradigm for advancing ViT-based video understanding models.

## 1 INTRODUCTION

Video understanding is central to a wide range of real-world applications, including autonomous driving (Dosovitskiy et al., 2017), surveillance (Chowdhury et al., 2021), sports analytics (Yan et al., 2019), and clinical training (Funke et al., 2022b). Unlike static image tasks, video understanding requires capturing not only what objects are present but also how they interact and evolve over time. Accurately distinguishing actions such as picking up versus putting down a tool or opening versus closing a door requires modeling temporal order, object interaction, and state changes (Sigurdsson et al., 2016; Damen et al., 2018). These capabilities are critical for safety-critical and decision-making systems (Zhu et al., 2020; Hu et al., 2021).

Despite significant progress with CNN- and Transformer-based video architectures (Carreira & Zisserman, 2017; Bertasius et al., 2021; Arnab et al., 2021), current models often fail to attend to the most relevant spatiotemporal cues (Stroud et al., 2020; Wu et al., 2022). Attention maps may highlight irrelevant background regions or static objects unrelated to the action, leading to unreliable predictions (Seo et al., 2022). Moreover, most models operate as black boxes, providing little insight into their decision-making process and lacking mechanisms to incorporate reasoning or correct spurious attention (Doshi-Velez & Kim, 2017; Gunning & Aha, 2019). This results in poor generalization under occlusion, viewpoint change, or distribution shift (Wang et al., 2022a; Li et al., 2023b).

Reasoning over object interactions, temporal order, and causal relations is essential for robust video understanding (Zellers et al., 2021; Huang et al., 2023). It allows models to disambiguate visually similar actions such as stirring versus pouring, understand action dependencies, and resist spurious correlations (Wu et al., 2021; Wang et al., 2022b). Incorporating reasoning improves interpretability and user trust by making predictions more faithful and consistent with human expectations (Goyal et al., 2017; Yi et al., 2020; Arrieta et al., 2020).

Spatial reasoning focuses model attention on task-relevant objects such as hands, tools, and manipulated objects while suppressing irrelevant regions, reducing distraction from background noise (Gao et al., 2023). Temporal reasoning captures how these objects change over time, enabling the model

to infer action directionality and temporal dependencies (Wang et al., 2021). For example, in surgical training videos, detecting whether a syringe is being filled or emptied requires reasoning over both object state and temporal relation across multiple frames (Funke et al., 2022b).

To be effective, spatial and temporal reasoning signals must be injected into the model during training rather than used only as post-hoc explanations, illustrated in Figure 1. This converts reasoning from an after-the-fact narrative into a learning signal that shapes internal representations. For instance, aligning attention maps with human-annotated object masks forces the model to focus on semantically meaningful regions, while enforcing temporal consistency across frames encourages smooth and coherent predictions over a clip. Together, these mechanisms improve both accuracy and interpretability, leading to more robust and trustworthy video understanding systems.

Performing Reasoning Supervision poses several unique challenges. **Generating training-time guidance** is a significant challenge: how to produce reliable guidance signals during training without human supervision. Automatically derived cues are often noisy, biased, or dataset-dependent, and they risk leaking label information or failing under distribution shift. **What specific guidance should be generated**, beyond the generation method, the semantic content of the guidance is critical. "Garbage in, garbage out". The injection of poorly chosen or irrelevant guidance can misalign the learning process with the intended objective and ultimately degrade model performance. **How to inject this guidance into the model**. This process requires careful selection of the injection target, method, and schedule. A suboptimal strategy can introduce several issues, such as over-regularizing the model, conflicting with the task objective, or destabilizing the optimization process.

We propose a Reasoning Supervision framework that integrates Vision Transformer (ViT) architectures with LLM-driven reasoning generation to enhance video understanding. Our framework is model-agnostic and can be applied to any ViT-based

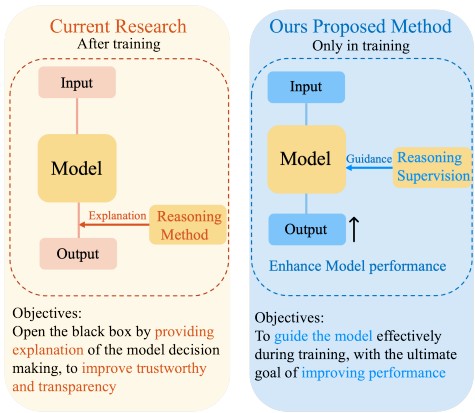

Figure 1: Comparison between conventional reasoning methods and our approach. Traditional reasoning techniques primarily perform post-hoc analysis to explain model decisions, whereas our method integrates reasoning signals into the training process, providing guidance that improves model performance.

video model. **First**, we leverage LLMs as proxy human annotators to generate training-time spatial guidance, producing human-free supervision signals. **Second**, we perform a task-aligned analysis to determine the optimal content and granularity of the guidance, identifying signals that improve model reasoning and robustness while distilling actionable insights for future research. **Third**, we introduce two complementary alignment objectives that inject spatial and temporal guidance into the model: a spatial alignment loss that aligns frame-level self-attention with LLM-derived spatial maps, and a temporal consistency loss that enforces agreement across three levels of temporal reasoning. Our experiments demonstrate that Reasoning Supervision consistently improves accuracy across ViT-based models.

## 2 REASONING SUPERVISION FRAMEWORK

The Reasoning Supervision framework comprises three key components: a Vision Transformer model, LLM-driven guidance generation, and reasoning guidance injection. The overall framework is depicted in Figure 2. To more effectively capture and enhance reasoning capabilities, we design a ViT architecture(TSViT) tailored for spatiotemporal reasoning. We further provide a detailed motivation for both the architectural choices and the LLM-driven reasoning guidance generation process, illustrating how they jointly supply structured, training-time guidance to improve model performance.

## 2.1 Vision Transformer Model

Our framework is designed to inject structured spatial and temporal signals into any model with spatial attention and temporal modeling capabilities. Models that produce attention maps can align them with external spatial guidance, while those with temporal representations can leverage consistency signals to improve long-range performance. In this study, we focus on Vision Transformer (ViT)–based architectures, which naturally yield token-level attention maps and temporal embeddings. We proposed a two-stage design, TSViT, where the first stage performs frame-level spatial learning and the second stage models temporal dependencies across frames. This structure enables the direct injection of LLM-derived spatial maps and temporal consistency, thereby aligning the model's internal representations with task-relevant evidence.

**Two-Stage Vision Transformer**. The design of TSViT is inspired by the way humans process visual information. When watching a video, we first recognize what is present — the objects, the scene, and the key spatial regions — and then reason about how events unfold over time. For example, distinguishing picking up a screwdriver from putting it down requires understanding not only the presence of the screwdriver but also the temporal information of events. Likewise, in autonomous driving, recognizing whether a pedestrian intends to cross the street requires looking at several consecutive frames rather than a single snapshot. Motivated by this reasoning process, TSViT is designed as a two-stage framework that first performs spatial learning at the frame level and then performs temporal learning at the clip level. The detailed structure is shown in Figure 3.

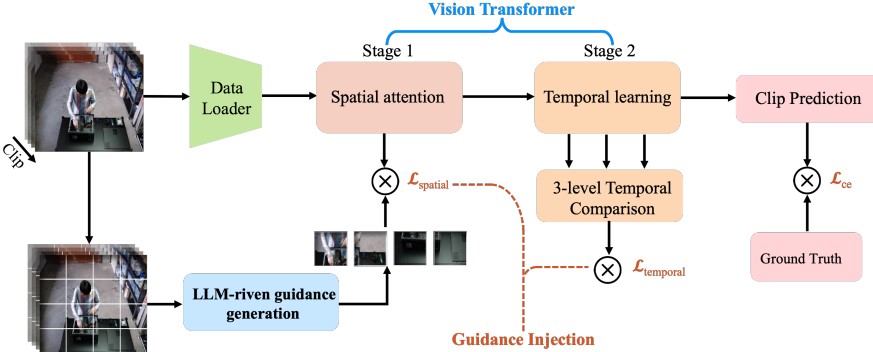

Figure 2: Overview of the proposed Reasoning Supervision framework. LLMs generate spatial knowledge that guides ViT via spatial alignment objectives and three-level temporally consistent learning, ensuring coherent spatiotemporal reasoning throughout training.

**Frame-Level Spatial Learning**. In the first stage, TSViT focuses on spatial understanding by learning which regions of each frame are most relevant for the downstream task. This is analogous to a human first scanning a scene to identify key objects and regions of interest. For instance, in the Desktop Assembly dataset Shi et al. (2020), the model should focus on the desktop and the user's hands, while keeps less attention in irrelevant regions such as the background or unused tools. This stage produces a set of frame-level representations that capture the most informative spatial features.

**Clip-Level Temporal Learning.** In the second stage, TSViT takes the frame-level representations as input tokens and models the temporal dynamics across the entire clip, rather than relying on isolated frames or short segments. To encourage temporally coherent reasoning, we introduce a hierarchical temporal consistency. Human actions naturally exhibit a temporal hierarchy, including immediate cues that describe instantaneous motion (e.g., hand moving), contextual dependencies over recent frames (e.g., reaching for an object), and a narrative view that captures the overall semantic goal (e.g., making coffee ). Our key insight is that predictions at these three levels should be mutually consistent, as they describe the same action at different temporal scales. We therefore penalize their discrepancies, encouraging the model to produce temporally aligned representations that respect the hierarchical structure of human action understanding.

## 2.2 LLM-Driven Knowledge Generation

Recently, Human-in-the-Loop (HITL) learning has demonstrated success across various domains, including the interactive annotation Settles (2011), active learning for autonomous perception

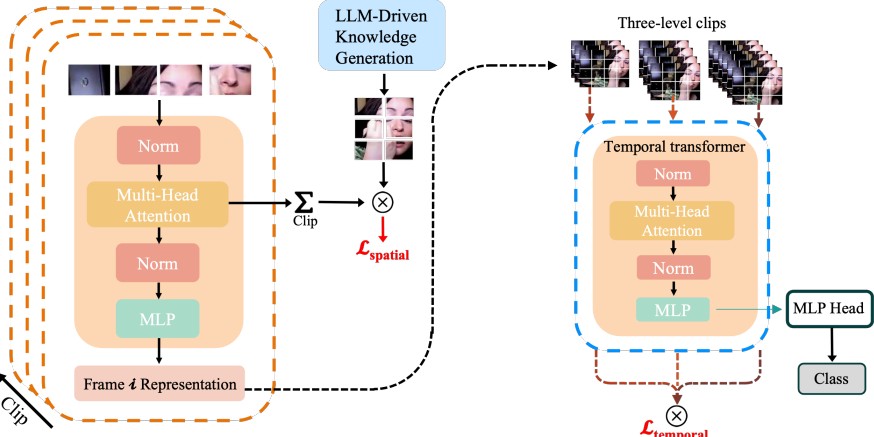

Figure 3: The framework of TSViT. The first stage performs spatial learning by taking patched frame embeddings as input and computing a spatial alignment loss. The second stage performs temporal learning, using the outputs from the first stage across a clip to compute a temporal alignment loss. Finally, the resulting representation is passed through an MLP head to predict the clip-level class.

Konyushkova et al. (2017), and reinforcement learning with human feedback for LLM alignment Christiano et al. (2017). Despite its effectiveness, HITL approaches require substantial human effort, time, and cost, which limit their scalability in domains where expert annotations are scarce or prohibitively expensive, such as the human activity recognition task, eg, surgical workflow recognition Funke et al. (2022a). Recent advances in large language models (LLMs) reveal strong emergent reasoning abilities, such as chain-of-thought prompting Wei et al. (2022b) and in-context learning Brown et al. (2020), enabling them to perform complex reasoning without direct human supervision. We posit that these capabilities can be leveraged to generate task-relevant knowledge automatically, reducing annotation cost and accelerating model development. However, a key question remains: how to systematically select the reasoning knowledge produced by LLMs so that it is both correct and beneficial for downstream training.

**Knowledge Selection**. Our approach is inspired by a self-distillation method, DINO Caron et al. (2021), where a teacher network learns from a global view of the input while a student network learns from local views, enabling the transfer of richer, more informative representations. The superior performance of the teacher model is expected, as it has access to more comprehensive information than the student. In a similar spirit, we aim to provide the model with non-trivial, complementary guidance that cannot be directly obtained from the dataset itself. Conventional data augmentation techniques (e.g., random cropping, rotation, flipping) Shorten & Khoshgoftaar (2019) merely produce alternative views of the same data and remain within the dataset domain. Thus, contribute little to the model's ability to acquire new, semantically meaningful representations. To overcome this limitation, we leverage LLMs to generate reasoning guidance. Specifically, our method uses LLMs to identify key objects in the frame and infer their spatial locations, effectively providing the model with explicit semantic cues. This form of knowledge goes beyond pixel-level augmentation by providing explicit reasoning signals, guiding models to focus on key objects and their spatial locations.

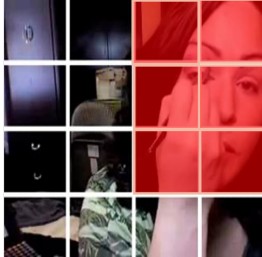

Figure 4: Example of the $4 \times 4$ grid representation applied to a frame from the UCF-101 dataset. The highlighted red regions denote task-relevant locations identified by the LLM-generated spatial guidance.

**Knowledge Generation**. In our proposed approach, we employ ChatGPT to generate reasoning outputs. While LLMs demonstrate strong reasoning abilities, they are not explicitly designed for object localization. Inspired by the recent work, Yang et al. (2024), which segments individual objects and feeds them to the LLM for grounding, we adopt a simpler yet effective strategy: we partition each frame into a $4 \times 4$ grid, resulting in the same number of tokens as used in our TSViT design. This grid-based representation provides a structured spatial decomposition of the frame, aligning natu-

rally with the ViT tokenization scheme and improving reasoning accuracy. An illustrative example from the UCF101 dataset Soomro et al. (2012) is shown in Figure 4.

## 2.3 REASONING GUIDANCE INJECTION

To effectively inject LLM-derived guidance into the model, we design two complementary loss functions that operate on the spatial and temporal dimensions of video representation learning. The spatial alignment loss explicitly aligns the model's attention maps with task-relevant regions identified by the LLM, ensuring that the network focuses on semantically meaningful areas rather than spurious background cues. The three-level temporal consistency loss, on the other hand, regularizes the model's predictions across multiple temporal scales, encouraging agreement between short-term, mid-range, and full-clip reasoning.

**Setup and Notation.** Let a video clip contain $T$ frames, each divided into a $g_h \times g_w$ grid, giving $N = g_h g_w$ patch tokens per frame. For a batch of size $B$, the spatial self–attention tensor at a chosen Transformer layer has shape $\mathbf{A} \in \mathbb{R}^{(BT) \times H_h \times (1+N) \times (1+N)}$, where $H_h$ is the number of attention heads and the first token corresponds to the `[CLS]` token. We extract the attention from `[CLS]` to all patch tokens and reduce across heads:

$$\hat{\mathbf{a}}_{b,t,h} = \mathbf{A}_{(b,t),h,0,1:N} \in \mathbb{R}^N, \qquad \tilde{\mathbf{a}}_{b,t} = \frac{1}{H_h} \sum_{h=1}^{H_h} \hat{\mathbf{a}}_{b,t,h}.$$

Normalizing over the $N$ patches gives a probability distribution

$$\mathbf{p}_{b,t} = \frac{\tilde{\mathbf{a}}_{b,t}}{\mathbf{1}^\top \tilde{\mathbf{a}}_{b,t} + \varepsilon} \in \Delta^{N-1},$$

where $\varepsilon$ is a small constant for stability. If the attention size $K \neq N$, we follow the implementation by padding $\mathbf{p}_{b,t}$ with zeros (if $K < N$) or truncating (if $K > N$) to ensure dimension consistency.

**LLM-Derived Supervision.** For each frame, we build a binary ground-truth mask $\mathbf{M}_{b,t} \in \{0,1\}^N$ from LLM annotations, and a frame-level indicator $F_{b,t}$ which is 1 if at least one token is annotated. Valid frames are normalized to form a distribution and frames with no positive tokens are skipped.

$$\mathbf{q}_{b,t} = \frac{\mathbf{M}_{b,t}}{\mathbf{1}^\top \mathbf{M}_{b,t}} \in \Delta^{N-1},$$

### 2.3.1 SPATIAL ALIGNMENT LOSS

The spatial alignment loss encourages the model to place high attention mass on annotated tokens. For each valid frame $(b,t)$, we compute the KL divergence between $\mathbf{q}_{b,t}$ and $\mathbf{p}_{b,t}$:

$$\mathcal{L}_{\text{spatial}} = \frac{1}{|\mathcal{V}|} \sum_{(b,t) \in \mathcal{V}} \mathrm{KL}\big(\mathbf{q}_{b,t} \,\|\, \mathbf{p}_{b,t}\big) = \frac{1}{|\mathcal{V}|} \sum_{(b,t) \in \mathcal{V}} \sum_{n=1}^{N} q_{b,t,n} \log \frac{q_{b,t,n}}{p_{b,t,n} + \varepsilon},$$

where $\mathcal{V} = \{(b,t) : F_{b,t} = 1\}$ is the set of frames with valid annotations. Following the implementation, frames with degenerate distributions (zero mass) are excluded, and the loss is averaged across all remaining frames.

### 2.3.2 THREE-LEVEL TEMPORAL CONSISTENCY LOSS

In the second stage, TSViT treats the $T$ frame-level representations as temporal tokens and models dependencies across the entire clip, capturing both short-term motion and long-range context. Rather than relying on isolated frames or short segments, this stage performs joint reasoning over the full temporal sequence. We explicitly consider three levels of temporal granularity: (1) *Immediate cues*, derived from the first $20\%$ of frames and capturing fine-grained motion, (2) *Contextual cues*, obtained from the middle $60\%$ of frames and encoding mid-range dependencies, and (3) *Narrative semantics*, computed over the entire clip to represent the global action intent. These three temporal views describe the same action from complementary perspectives and should therefore produce mutually consistent predictions.

**Formulation.** For each sample $b$, let $p_b^{(\mathrm{im})}$, $p_b^{(\mathrm{ctx})}$, and $p_b^{(\mathrm{nar})} \in \Delta^{C-1}$ denote the class probability distributions obtained from the immediate, contextual, and narrative temporal segments, respectively, after temperature-scaled softmax:

$$p_b^{(l)} = \mathrm{softmax}\!\left(\frac{\mathbf{z}_b^{(l)}}{\tau}\right), \qquad l \in \{\mathrm{im}, \mathrm{ctx}, \mathrm{nar}\},$$

where $\tau$ is a temperature parameter. We compute a consensus distribution

$$\bar{p}_b = \frac{1}{3}\big(p_b^{(\mathrm{im})} + p_b^{(\mathrm{ctx})} + p_b^{(\mathrm{nar})}\big),$$

which represents the shared prediction across temporal scales. The three-level temporal consistency loss is defined as the Jensen–Shannon divergence:

$$\mathcal{L}_{\mathrm{temporal}} = \frac{1}{|\mathcal{B}|}\sum_{b \in \mathcal{B}} \frac{1}{3}\Big[\mathrm{KL}\big(p_b^{(\mathrm{im})} \,\|\, \bar{p}_b\big) + \mathrm{KL}\big(p_b^{(\mathrm{ctx})} \,\|\, \bar{p}_b\big) + \mathrm{KL}\big(p_b^{(\mathrm{nar})} \,\|\, \bar{p}_b\big)\Big].$$

**Interpretation.** This objective encourages each temporal view to agree with the consensus distribution, aligning short-term, mid-range, and full-clip predictions. By enforcing this multi-scale agreement, TSViT learns temporally coherent representations that respect the hierarchical structure of human action understanding and produce more stable and robust action predictions.

**Overall Loss Function.** The final training objective combines standard classification loss with the proposed spatial and temporal alignment terms:

$$\mathcal{L}_{\mathrm{total}} = \mathcal{L}_{\mathrm{CE}} + \lambda_s \mathcal{L}_{\mathrm{spatial}} + \lambda_t \mathcal{L}_{\mathrm{temporal}},$$

where $\mathcal{L}_{\mathrm{CE}}$ is the conventional cross-entropy loss between the model's predicted class probabilities and the ground-truth label. The coefficients $\lambda_s$ and $\lambda_t$ control the relative contribution of the spatial alignment and temporal consistency terms. This formulation jointly optimizes recognition accuracy and interpretability by encouraging the model to focus on semantically meaningful regions and maintain temporally coherent predictions.

## 3 Reasoning Supervision for Human Activity Recognition

In this section, we evaluate the proposed framework on several human activity recognition benchmarks. Our experiments are designed to answer the following research questions: (1) *To what extent does Reasoning Supervision improve the performance of Vision Transformer–based models?* (2) *How do the spatial alignment and temporal consistency losses influence model training and representation learning?*

### 3.1 Experiment Setup

Our experimental setup includes a large-scale dataset (SSv2 Goyal et al. (2017)), a medium-scale dataset (UCF101 Soomro et al. (2012)), a small-scale dataset (UCF50 Reddy & Shah (2013)), and a domain-specific dataset (Desktop Assembly Yuan & et al. (2024)). For comparison, we adopt state-of-the-art transformer-based baselines, including ViViT Arnab et al. (2021), TSViT, TimeSformer Bertasius et al. (2021), and its pretrained version. We report standard metrics for the human activity recognition task, including Top-1 accuracy, Top-5 accuracy, and clip-level loss. For the Desktop Assembly dataset, we additionally evaluate using segmentation-specific metrics: Top-1 accuracy, clip loss, mean-over-frames (MoF), edit distance, and F1 scores at 10% and 25% overlap thresholds (F1@10, F1@25, F1@50).

For efficiency, we precompute the reasoning guidance prior to training. Specifically, we sample 60% of the training videos from UCF50 and UCF101, extracting 10 uniformly spaced frames per video to generate spatial guidance. Since these datasets consist of short clips, this provides sufficient coverage. For Desktop Assembly, we process all videos at 10 fps due to its smaller scale, while for the much larger dataset, SSV2, we sample only 10% of the training videos to reduce cost. After obtaining LLM outputs, we aggregate them into spatial distribution maps and precompute per-clip spatial and temporal losses to accelerate training. All reasoning signals are generated using GPT-4.1-mini (OpenAI, 2024)

## 3.2 Performance of Reasoning Supervision Framework(RQ1)

Table 1: Clip-level result comparison of each baseline and TSViT with vs. without Reasoning Supervision (RS). An asterisk (*) indicates pretraining on Kinetics-400(K400) datasets as reported in the original papers. Higher is better for accuracy (↑); lower is better for loss (↓).

| Dataset | Method | Top-1 Acc. ↑ | Top-5 Acc. ↑ | Clip Loss ↓ |
|---------|--------|--------------|--------------|-------------|
| SSv2 | TimeSformer* | 59.13 | **85.47** | 1.589 |
| | TimeSformer* + **RS** | **59.77** +1.082% | 84.21 -1.474% | **1.526** -0.63 |
| UCF101 | TimeSformer* | 98.79 | **99.41** | 0.2172 |
| | TimeSformer* + **RS** | **99.40** +0.617% | 99.02 -0.392% | **0.2170** -0.0002 |
| | TimeSformer | 82.41 | 94.22 | 0.6950 |
| | TimeSformer + **RS** | **85.36** +3.77% | **94.82** +0.637% | **0.6278** -0.0672 |
| | ViViT | 68.47 | 84.83 | 1.517 |
| | ViViT + **RS** | **71.87** +4.97% | **86.13** +1.53% | **1.474** -0.043 |
| | TSViT | 67.62 | 85.66 | 1.571 |
| | TSViT + **RS** | **70.63** +4.45% | **88.15** +2.91% | **1.421** -0.15 |
| UCF50 | TimeSformer* | 96.56 | 98.88 | 0.266 |
| | TimeSformer* + RS | **99.18** +2.71% | **99.93** +0.303% | **0.257** -0.09 |
| | TimeSformer | 67.23 | 84.61 | 1.483 |
| | TimeSformer + **RS** | **88.93** +32.3% | **97.31** +15.0% | **0.434** -1.049 |
| | ViViT | 71.50 | 90.73 | 1.0974 |
| | ViViT + **RS** | **74.37** +4.13% | **91.27** +0.595% | **1.0648** -0.0326% |
| | TSViT | 75.14 | 87.62 | 1.525 |
| | TSViT + **RS** | **80.07** +6.56% | **91.13** +4.01% | **1.157** -0.368 |

**Reasoning Supervision improves accuracy in ViTs**. On the UCF50 dataset, all baselines benefit from Reasoning Supervision, with particularly large gains for TimeSformer, as shown in Table 1. Its Top-1 accuracy increases from 67.23% to 88.93%, a relative improvement of over 32%, while Top-5 accuracy improves by 15%. The clip-level loss drops dramatically from 1.483 to 0.434, highlighting the effectiveness of our framework. These results demonstrate that Reasoning Supervision can be especially valuable when training data is limited. TimeSformer has over 88M parameters, making it a relatively large model for human activity recognition, and ViTs are known to be data-hungry. UCF50 contains only 50 classes and fewer than 7,000 videos, which is insufficient to fully exploit such a large model. Without reasoning supervision, TimeSformer achieves only 67% Top-1 accuracy, whereas with Reasoning Supervision it approaches 90%. We attribute this gain to the additional spatial and temporal guidance provided by Reasoning Supervision, which injects task-relevant information beyond what is available in the raw dataset and improves both learning efficiency and generalization.

On the UCF101 dataset, we observe consistent but smaller improvements from Reasoning Supervision compared to UCF50, which is expected given the larger scale and diversity of UCF101. With sufficient training data, large Vision Transformer models can leverage their capacity more effectively: for instance, the non-pretrained TimeSformer achieves over 82% Top-1 accuracy, while its pretrained counterpart surpasses 99%. In this high-performance regime, Reasoning Supervision yields only marginal gains, and for the pretrained TimeSformer, Top-5 accuracy slightly decreases. These findings suggest that Reasoning Supervision is most beneficial in data-limited settings or for under-parameterized models, rather than for fully pretrained, highly accurate architectures. We also evaluate ViViT, a compact ViT model with 4.3M parameters. On UCF101, ViViT attains just above 70% Top-1 accuracy, but incorporating Reasoning Supervision yields nearly a 5% absolute improvement and also reduces clip-level loss. This result highlights the effectiveness of our framework in enhancing the learning of lightweight, non-pretrained models by injecting spatial and temporal reasoning signals that are otherwise difficult to acquire from limited data.

SSv2 poses a significant challenge due to its diverse activities and high environmental variability, which increase classification difficulty. To ensure a fair evaluation and reduce underfitting, we report results only with the pretrained TimeSformer. The observed improvement is marginal, likely because the spatial reasoning guidance is too sparse to significantly influence model performance. Since we

generate guidance for only 10% of the training data and SSv2 contains many classes, the model may not receive sufficient coverage to reliably learn task-relevant spatial guidance.

We further evaluate the effectiveness of Reasoning Supervision in videos containing multiple sequential actions, with results reported in Table 2. On the Desktop Assembly dataset, TSViT shows significant improvements across all metrics, indicating that Reasoning Supervision is particularly effective in scenarios with rapid action transitions and fine-grained temporal dependencies.

Table 2: Performance on the Desktop Assembly Benchmark.

| Method | Top-1 Acc.↑ | Clip Loss↓ | MoF↑ | Edit↑ | F1@10↑ | F1@25↑ |
|---|---|---|---|---|---|---|
| TSViT | 72.91 | 1.1324 | 67.86 | 65.98 | 76.16 | 71.51 |
| TSViT + **RS** | **82.78** | **0.8116** | **76.50** | **80.41** | **83.94** | **82.82** |
| | +13.53% | -0.3208 | +12.73% | +21.87% | +10.22% | +15.82% |

### 3.3 LOSS DYNAMICS DURING TRAINING

We analyze the loss curves of a non-pretrained TimeSformer trained on UCF101 with Reasoning Supervision (Figure 5). The spatial alignment loss decreases steadily, indicating that the model progressively focuses on task-relevant regions and aligns its attention with the LLM-derived guidance. This reduction correlates with improved classification accuracy, confirming that better spatial focus enhances recognition. The temporal consistency loss remains low and stable, suggesting that predictions across immediate, contextual, and full-clip views are well aligned throughout training. Overall, the monotonic decrease in total loss and increase in accuracy demonstrate that Reasoning Supervision effectively guides the model to learn spatial and temporal reasoning, resulting in improved performance.

## 4 RELATED WORK

**Vision Transformers for Video Understanding.** Transformer-based video models such as ViViT (Arnab et al., 2021), TimeSformer (Bertasius et al., 2021), and MViT (Fan et al., 2021) achieve state-of-the-art performance by modeling spatiotemporal dependencies using self-attention. ViViT explores multiple factorization schemes (space–time joint, factorized, and tubelet) but relies solely on classification supervision. Our proposed TSViT differs in two key ways: (1) it decouples spatial and temporal reasoning into two explicit stages, and (2) it introduces spatial alignment and three-level temporal consistency losses that regularize attention to be semantically meaningful and temporally coherent, turning reasoning into a first-class training signal rather than a byproduct.

**Explainable AI and Reasoning Supervision.** Explainable AI (XAI) techniques such as saliency-based attribution (Ancona et al., 2018; Zhu et al., 2024), path-based methods (Zhang

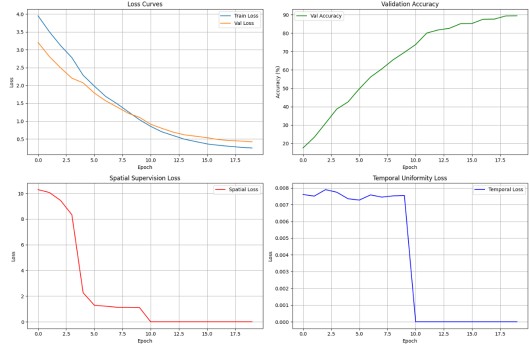

Figure 5: Training dynamics of TimeSformer on the UCF101 dataset. The top-left panel shows the overall training loss, the top-right shows the accuracy curve, the bottom-left shows the spatial alignment loss, and the bottom-right shows the temporal consistency loss. To better show the result, we only plot the first 18 epoches.

et al., 2024), and concept-level explanations (Sun et al., 2025) aim to reveal the evidence behind model decisions. However, most remain post-hoc and do not influence model training, limiting their effect on robustness. Recent work has begun to use reasoning supervision—aligning model attention with human annotations or proxy cues—to improve performance and interpretability (Wang et al., 2023; Li et al., 2023a). Our framework builds on this idea by injecting reasoning signals throughout training, jointly optimizing task accuracy and interpretability.

**LLM-Guided Knowledge Distillation.** Large language models (LLMs) have recently been used to generate pseudo-labels, explanations, and fine-grained supervision signals (Wei et al., 2022a; Liu et al., 2023; Li et al., 2023a; Wang et al., 2023). Unlike direct prompt-based inference, LLM-guided distillation leverages these outputs to supervise smaller models during training, reducing annotation costs while maintaining strong performance. In our work, we exploit LLMs as surrogate annotators to provide spatial maps and temporal cues, enabling human-free reasoning supervision for video understanding.

## 5 LIMITATIONS & FUTURE DISCUSSION

While Reasoning Supervision improves performance across multiple video benchmarks, its effectiveness is inherently tied to the quality, stability, and consistency of the underlying LLM used to generate reasoning guidance. LLM outputs may contain noise or bias, which could propagate into the supervision signal and affect model training. Moreover, generating spatial maps and temporal cues for large-scale datasets introduces additional monetary cost (see Appendix for detailed analysis), which can limit adoption in resource-constrained or real-time scenarios. Future work will explore strategies to reduce these costs, including caching and reusing reasoning traces across similar videos, performing batched or parallelized inference, and distilling LLMs into lightweight student models for more efficient guidance generation. Another promising direction is to generate richer and more structured temporal reasoning cues, such as event boundaries or causal chains, directly from LLM outputs and integrate them into temporal learning objectives.

## 6 SUMMARY

In this work, we introduced Reasoning Supervision, a training paradigm that injects spatial and temporal reasoning signals into Vision Transformer models for human activity recognition. Our framework leverages LLM-derived spatial maps and multi-scale temporal cues, aligning the model's internal attention distributions with task-relevant evidence and enforcing temporal consistency across clips. Extensive experiments across multiple benchmarks, including SSv2, UCF101, UCF50, and the Desktop Assembly dataset, demonstrate that Reasoning Supervision consistently improves performance. Notably, we observed substantial gains on data-scarce benchmarks such as UCF50, where Top-1 accuracy improved by more than 32%, and on multi-action scenarios in Desktop Assembly, where improvements were seen across MoF, Edit, and F1 metrics. Our analysis of training dynamics further revealed that spatial alignment loss decreases steadily while accuracy increases, confirming that Reasoning Supervision encourages models to focus on semantically meaningful regions and maintain temporally coherent predictions.

## ETHICS STATEMENT

This work uses publicly available benchmark datasets (SSv2, UCF101, UCF50, Desktop Assembly) that contain non-identifiable human activity videos and comply with their respective licenses. No new human data were collected. Our approach leverages LLM-generated reasoning signals, which may reflect biases present in the LLM training data and potentially influence model behavior. We mitigate these risks through diverse datasets and attention-map analysis to ensure focus on task-relevant regions. While our method improves transparency and accuracy, we caution against misuse in privacy-sensitive contexts and encourage further research on fairness and privacy-preserving reasoning before deployment in safety-critical applications.

## REPRODUCIBILITY STATEMENT

To ensure reproducibility, we include detailed implementation in the Appendix. The exact LLM prompts used to generate spatial reasoning guidance are also provided, together with representative examples of model outputs. We will publicly release the full set of LLM-generated spatial reasoning guidance for UCF50 and UCF101, as well as the complete codebase, to facilitate further research and benchmarking. All code and data will be made available at https://github.com/asdbfioioiyuf/Reasoning-Supervision in a week after submission.

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
