## A  THE USE OF LLM

In this work, large language models (LLMs) are used in two primary ways. First, we employ an LLM to improve the readability and grammar of the manuscript, ensuring that the writing is clear and accessible to a broad research audience. Second, and most critically, we rely on LLMs to generate spatial reasoning guidance for our proposed Reasoning Supervision framework. For each dataset, we provide carefully designed prompts to extract object-level spatial maps and temporal cues that highlight task-relevant regions and dependencies. The exact prompts and representative outputs are included in the Appendix to ensure reproducibility. All reasoning signals were generated using `GPT-4.1-mini`, which was selected for its cost-efficiency and strong reasoning capability.

## B  LLM PROMPTS AND COST

This section provides the exact prompts used for generating spatial reasoning guidance to ensure reproducibility.

The approximate token cost for generating spatial reasoning guidance for a single $320 \times 240$ frame in the UCF datasets is about 338 tokens. Each video contains 10 sampled frames, resulting in roughly 3380 tokens per video. For UCF50 and UCF101, where we process 60% of the training videos, this yields approximately $0.6 \times N_{\text{train}} \times 3380$ total tokens (where $N_{\text{train}}$ is the number of training videos). For SSv2, the per-frame token cost is slightly higher at around 360 tokens due to longer class labels. Since we process 10% of the training set with 10 frames per video, the total token count is approximately $0.1 \times N_{\text{train}} \times 3600$. For Desktop Assembly, the per-frame cost ranges from 450–500 tokens, reflecting the additional object-level annotations required in its reasoning outputs. The example outputs from LLMs are shown below.

**Desktop Response**

```
"important_objects": [ "chip", "hand" ],
"localization": {
              "chip": [ [2, 2 ]],
              "hand": [ [ 1, 2 ] ]}
```

Figure 1: Output for DA dataset

**UCF101 Response**

```
"v_BaseballPitch_g08_c05.avi": {
                    "0": {
                            "text": "{ \"key_r\": [[1, 1], [2, 1], [2, 2]] }"
                    }
```

**SSV2 Response**

```
"78687.webm": {
        "0": {
        "text": "{ \"key_r\": [[2, 1], [2, 2], [3, 1], [3, 2]] }"
        }
```

Figure 2: Output for UCF and SSV2 dataset

**System message**

You are an expert in human activity recognition, with a strong focus on identifying key regions within video frames that are most relevant to the action being performed. Your deep knowledge of the UCF101dataset allows you to distinguish task-relevant objects and actions.

**Prompt message**

You are given a single frame from a video along with its ground truth action label. The frame is conceptually divided into a 4×4 grid: 4 rows (indexed 0 to 3 from top to bottom) 4 columns (indexed 0 to 3 from left to right)

Your task is to analyze the frame and complete the following:
      For each frame, output the following:
      Key Region Selection: Identify the grid cell(s) that are most relevant to the given action. Return them as a list of     (row, column) pairs, where each pair corresponds to a grid cell index.

OUTPUT FORMAT: {{ "key_r": [[row1, col1], [row2, col2], ...] }}

Figure 3: Prompt for UCF dataset

**System message**

You are an expert in human activity recognition, with a strong focus on identifying  key regions within video frames that are most relevant to the action being performed. Your deep knowledge of the SSV2 dataset allows you to distinguish task-relevant objects and actions.

**Prompt message**

You are given a single frame from a video along with its ground truth action label.
The frame is conceptually divided into a 4×4 grid:
4 rows (indexed 0 to 3 from top to bottom)
4 columns (indexed 0 to 3 from left to right)
Your task is to analyze the frame and complete the following:
INSTRUCTIONS: For each frame, output the following:

Key Region Selection: Identify the grid cell(s) that are most relevant to the
given action. Return them as a list of (row, column) pairs, where each pair
corresponds to a grid cell index.

Figure 4: Prompt for SSV2 dataset

Figure 5: Prompt for Desktop Assembly dataset

## C  EXTENDED RELATED WORK

**Vision Transformers for Video Understanding.**  The introduction of Vision Transformers (ViTs) has significantly advanced video understanding by enabling global spatiotemporal modeling through self-attention mechanisms. Models such as ViViT (Arnab et al., 2021), TimeSformer (Bertasius et al., 2021), and MViT (Fan et al., 2021) represent milestone architectures in this direction. ViViT systematically explores different factorization strategies for attention computation, including joint space–time attention, factorized attention (where spatial and temporal dimensions are processed separately), and tubelet embeddings that reduce input sequence length. TimeSformer further demonstrates that factorized attention not only reduces computational cost but can also maintain or exceed the performance of joint attention on large-scale datasets such as Kinetics-400. Multiscale Vision Transformers (MViT) extend this paradigm by progressively reducing temporal and spatial resolution across layers, enabling computation-efficient yet powerful hierarchical feature representations.

Despite these advances, most existing models rely solely on classification supervision and lack explicit mechanisms to enforce semantically meaningful attention. As a result, attention maps may highlight irrelevant background regions or static objects, which can degrade robustness under occlusion, viewpoint changes, or domain shift (Wu et al., 2022; Seo et al., 2022). Our proposed TSViT is designed to address this limitation by decoupling spatial and temporal reasoning into two explicit stages: a frame-level spatial learning stage that focuses on task-relevant objects and regions, and a temporal reasoning stage that models multi-scale temporal dependencies. Crucially, TSViT is trained with complementary alignment objectives: a spatial alignment loss that constrains attention to align with LLM-derived guidance and a three-level temporal consistency loss that regularizes predictions across short-term, mid-range, and global temporal scales. This turns reasoning from a post-hoc interpretability tool into a first-class training signal that directly shapes model representations.

**Explainable AI and Reasoning Supervision.**  Explainable AI (XAI) aims to make model predictions more transparent by attributing decisions to input features or higher-level concepts. Classical attribution techniques include gradient-based saliency maps (Ancona et al., 2018), path-integrated gradients (Sundararajan et al., 2017; Zhang et al., 2024), and parameter-aware attribution for robust saliency (Zhu et al., 2024). Concept bottleneck models (Sun et al., 2025) and prototype-based networks (Chen et al., 2019) go further by introducing inherently interpretable intermediate representations. However, these methods are often used post-hoc or focus on interpretability at inference time, with limited influence on the training dynamics or robustness of the learned representations (Adebayo et al., 2018).

Recent research has introduced the notion of *reasoning supervision*, where attention or intermediate representations are explicitly aligned with human annotations, bounding boxes, or other proxy signals during training (Selvaraju et al., 2021; Liu et al., 2022). For example, CAM-guided loss functions have been used to suppress spurious correlations by penalizing attention outside labeled regions. More recent work leverages LLMs to generate pseudo-rationales for text–image pairs and uses these to regularize cross-modal attention (Wang et al., 2023; Li et al., 2023). Our framework generalizes this idea to video understanding, using LLM-generated spatial masks and temporal cues as supervision signals to improve both interpretability and performance.

**LLM-Guided Knowledge Distillation.** Large language models (LLMs) have demonstrated impressive reasoning and few-shot generalization capabilities, making them attractive sources of auxiliary supervision. Chain-of-thought prompting (Wei et al., 2022) and visual-language prompting (Liu et al., 2023) have been used to generate step-by-step rationales or scene descriptions, which can then be used to train smaller models in a distillation framework (Magister et al., 2022). Methods such as BLIP-2 (Li et al., 2023) and ChatGPT4V-based supervision (Wang et al., 2023) demonstrate that leveraging LLM-generated captions, explanations, or visual grounding signals can significantly improve model generalization while reducing the need for manual annotation.

Our work extends this paradigm by using LLMs as surrogate annotators to generate structured reasoning guidance at both the spatial and temporal levels. Unlike direct captioning or zero-shot prediction, we transform the LLM output into token-level supervision maps and temporal consistency objectives that are differentiable and can be directly integrated into model training. This approach converts LLM knowledge into a cost-effective, automated source of training-time guidance, bridging the gap between post-hoc explanation and end-to-end learning.