# OpenReview forum: "Reasoning Supervision for Vision Transformers in Human Activity Recognition"
_ICLR.cc/2026/Conference — ICLR 2026 Conference Withdrawn Submission_

### Official Review · Reviewer_neTy · 2025-10-18

**Soundness:** 2
**Presentation:** 2
**Contribution:** 2
**Rating:** 2
**Confidence:** 5

**Summary:**

This paper proposes a training framework named Reasoning Supervision (RS) to enhance the performance and interpretability of Vision Transformers (ViTs) in video action recognition tasks. The authors contend that most current "reasoning" methods rely on post-hoc interpretation. In contrast, this work aims to directly introduce "reasoning supervision signals" during the training phase to guide the model's internal attention distribution and temporal consistency.

Experiments conducted on several video datasets, including SSv2, UCF50, UCF101, and Desktop Assembly, demonstrate that RS significantly improves model performance (particularly on smaller datasets) while also enhancing its robustness and interpretability.

**Strengths:**

The proposed framework is model-agnostic and can be adapted to any Vision Transformer (ViT) architecture. It demonstrates effectiveness, notably in few-shot learning tasks on datasets such as UCF50 and Desktop Assembly, where it achieves substantial performance gains (up to +32% in Top-1 accuracy).

**Weaknesses:**

I find that the novelty of this paper is somewhat limited. Furthermore, I have several concerns regarding its methodological design and experimental setup:

- **Use of LLM for Guidance?** The paper mentions employing GPT-4.1-mini as an annotator to provide reasoning guidance. In my view, it would be more accurate to describe this as using a Visual Language Model (VLM) rather than a pure Large Language Model (LLM), as VLMs possess visual perception and reasoning capabilities. This implies that the reasoning supervision signals are distilled from a visual-language model, not a pure text-based one. Given that VLMs already exhibit visual reasoning ability and strong text alignment, why not directly use an open-source VLM as the starting point for training? VLMs have become a common paradigm in video tasks. Is there a specific advantage to the current approach, compared to using a VLM alone to generate textual guidance?

- **Reliability of VLM-Generated Signals:** While the authors acknowledge the presence of noise and bias in the LLM-generated outputs, there is no quantitative evaluation (such as a correlation analysis between signal quality and model performance) to assess the impact of these issues.

- **Lack of Experiments on Larger-Scale Datasets:** The experiments are conducted only on relatively small datasets. It remains unclear whether the proposed method generalizes well to larger and more representative benchmarks such as Kinetics-400, 600, or 700, all of which provide standard training and test splits. Evidence on such datasets is necessary to demonstrate the universality and scalability of the approach.

- **Insufficient Comparison with Other Reasoning-Based Methods:** The experimental comparisons focus mainly on traditional models, with no direct ablation or comparison against other reasoning-oriented or attention-alignment methods (e.g., those based on CLIP or other VLMs). This limits the claim of superiority in reasoning-aware modeling.

- **Lack of Visual Validation:** The claim of improved interpretability would be better supported if more visual examples were provided—for instance, by comparing attention maps before and after applying the proposed reasoning supervision.

**Questions:**

See Weakness.

In addition, the meaning of the variable *z* in line 274 is unclear. It should be clarified whether *z* represents the [CLS] token or what?. I could not find its definition in the main text.

The rationale behind the specific ratios assigned to the components ({im}→20%, {ctx}→60%, {nar}→100%) lacks sufficient justification. Is these values base on intuition? A comprehensive ablation study may necessary to empirically validate this configuration.

---

### Official Review · Reviewer_tZ4f · 2025-10-24

**Soundness:** 3
**Presentation:** 3
**Contribution:** 3
**Rating:** 6
**Confidence:** 3

**Summary:**

This paper introduces a brand new pipeline, the Reasoning Supervision framework, which moves beyond post-hoc explanation and is directly injected into the training stage. Starting with a comparison between conventional reasoning methods and the proposed approach, authors perform three significant challenges that their methodology stands for. Applied to ViT architectures, Reasoning Supervision can improve performance, establishing a simple yet effective paradigm for advancing ViT-based video understanding models. It introduces spatial alignment and three-level temporal consistency losses that regularize attention to be semantically meaningful and temporally coherent, turning reasoning into a first-class training signal rather than a byproduct.

**Strengths:**

Two-Stage Training Pipeline. TSViT is designed as a two-stage framework that first performs spatial learning at the frame level and then performs temporal learning at the clip level.
Unify among Different Temporal Scales. The TSViT is encouraged to produce temporally aligned representations among the hierarchical structure.
Leverage the Spatial Information. Detecting from a single frame, the attention map should be a guidance that reveals the most relevant area.
Detailed formulas. Authors provide sufficient evidence of the proposed framework and complementary loss.

**Weaknesses:**

Confused about "LLM-Driven Knowledge Generation": Strictly speaking, ChatGPT should not be just considered as LLMs, especially GPT-4o.  So the expression may lead to confusion.
No association information: And in subsection 2.2, "Knowledge Selection," paragraph, I think there is no relation between DINO and the proposed data generation way.
Incorrect cause and effect. In 2.3 Reasoning Guidance Injection, the LLM-derived guidance is only relevant to the spatial information, according to the formal paragraph. So the reason for two complementary losses should be reconsidered.
Ablation Study. Given the performance of the proposed framework among datasets, there should be ablation studies to demonstrate the importance of each component in the pipeline.

**Questions:**

See weakness

---

### Official Review · Reviewer_o1Jc · 2025-11-01

**Soundness:** 2
**Presentation:** 2
**Contribution:** 2
**Rating:** 4
**Confidence:** 4

**Summary:**

The paper introduces Reasoning Supervision, a training paradigm for vision transformers in human activity recognition that injects spatial and temporal reasoning signals during training. It leverages large language models to generate spatial guidance and enforces temporal consistency through complementary loss functions. Evaluation is performed on SSv2, UCF50 and UCF101 datasets.

**Strengths:**

* The proposed method tackles the important problem of integrating spatial and temporal cues for video classification.
* The proposed method shows improved performance in limited settings: e.g., on the UCF50 and UCF101 datasets with limited training data.
* The paper explains the proposed method in detail.

**Weaknesses:**

* The method does not seem to be applicable to state of the art methods such as Video-MAE, which outperforms the proposed method significantly. Video-MAE achieves 75.4% Top-1 accuracy on SSv2 versus only 59.77% Top-1 accuracy on SSv2 of the proposed method.

* The proposed method does not lead to significantly improved performance when the base models are already pre-trained K400. On both SSv2 and UCF101, the performance improvement on Top-1 accuracy is <= 1%. So training on a larger dataset negates the need of the proposed approach.

* The method uses a temporal consistency loss, with three levels of temporal granularity over immediate cues, derived from the first 20% of frames and contextual cues, obtained from the middle 60% of frames. These numbers 20% and 60% seem to be chosen arbitrarily and should be ablated.

* The paper should include additional datasets for evaluation such as "Ava: A video dataset of spatio-temporally localized atomic visual actions. CVPR, 2018.".  Following prior work such as Video-MAE.

* The paper should discuss prior work on grounding to fine-grained visual information in videos such as: "Look, Remember and Reason: Grounded reasoning in videos with language models, ICLR 2024"; "Fine-grained Spatiotemporal Grounding on Egocentric Videos, ICCV 2025".

* Typos: "LLM-riven" in Figure 2.

**Questions:**

* Does the proposed approach extend to state of the art models such as Video-MAE?
* The rationale behind selecting the 20& and 20% thresholds should be discussed in more detail.
* The paper should explain the choice of evaluation datasets in more detail.

---

### Official Review · Reviewer_Hnqe · 2025-11-05

**Soundness:** 2
**Presentation:** 2
**Contribution:** 2
**Rating:** 2
**Confidence:** 4

**Summary:**

This paper introduces a Reasoning Supervision framework that integrates spatial and temporal reasoning signals into Vision Transformers for human activity recognition. Unlike post-hoc explanation methods, it trains models to reason during learning using LLM-generated spatial guidance to highlight task-relevant regions without human labels. A custom two-stage ViT architecture (TSViT) handles per-frame spatial tokenization and temporal modeling. Two loss functions guide training: a spatial alignment loss aligning attention maps with LLM-identified regions, and a temporal consistency loss enforcing prediction coherence across time scales. This approach helps the model focus on semantically meaningful elements and reason more consistently over time. The contributions include LLM-driven supervision, multi-scale temporal alignment, and improved performance and interpretability across benchmarks.

**Strengths:**

- Significance - This work addresses an important challenge in video understanding – how to make models both more accurate and interpretable by injecting domain knowledge.
- Reasoning Guidance Injection - The concept of multi-scale temporal reasoning (immediate, contextual, narrative levels) is interesting in video action recognition, providing a hierarchical way to enforce prediction consistency.
- Undertrained Setting - It can be gleaned from Table 1 results that the method is more effective in undertrained settings, where reasoning supervision offers strong inductive bias and boosts performance.

**Weaknesses:**

- Scalability to Large Datasets - While the method seems to perform well on small and mid-scale datasets, its impact on the large and diverse Something-Something v2 dataset was minimal. This is likely due to sparse LLM-generated spatial cues, suggesting the approach may struggle in complex, high-variation settings without denser or more targeted supervision.
- LLM Dependency - The method depends heavily on LLM-generated supervision, raising concerns about guidance quality and computational cost. Inaccurate or biased LLM outputs could mislead the model, but the paper doesn't assess how often this occurs. Additionally, generating supervision for all frames is costly, forcing the authors to subsample frames, which may limit effectiveness.
- Ablation and Analysis - The paper lacks clear ablation to isolate the contributions of spatial and temporal losses. While qualitative insights are offered, quantitative comparisons (e.g., using only one loss at a time) are missing. Design choices like the 20%/60%/100% temporal split and loss weightings (λ_s, λ_t) are also not justified or tested for sensitivity, leaving questions about robustness and optimality.
- TSViT Impact - The benefits of TSViT are unclear. Without reasoning supervision, it performs similarly or slightly worse than standard baselines, and with supervision, other models still outperform it in Table 1. Strong results on Desktop Assembly are noted, but lack of comparison limits interpretation.
- Interpretability - Although not a major weakness, the paper emphasizes quantitative gains and alignment loss as a proxy for interpretability but lacks qualitative analysis. Visualizations comparing attention maps with and without reasoning supervision or user studies could better support authors' assertion of method leading to "semantically meaningful" representation. Without such examples or failure case analysis, the interpretability and robustness remain somewhat speculative.

**Questions:**

- Scalability to SSv2 -  Could denser or more targeted LLM guidance improve performance on large, diverse datasets like SSv2? Have you tested this beyond the current 10% annotation rate?
- LLM Guidance Quality - Did you assess how often LLM-generated masks are noisy or misleading? How robust is the model to imperfect guidance?
- Ablation Clarity - Can you provide quantitative results isolating the impact of spatial vs. temporal losses (and its three levels)? How do performance and robustness change when each is used independently?
- Design Sensitivity - Were the 20%/60%/100% temporal splits and loss weights (λ_s, λ_t) chosen empirically or tuned? Have you evaluated sensitivity to these choices?
- TSViT Justification - What unique benefits does TSViT offer compared to standard ViT backbones when used with reasoning supervision? Could you share results from Desktop Assembly using other architectures for comparison?
- Interpretability Evidence - Can you provide qualitative examples (e.g., attention maps or visualizations) showing that reasoning supervision improves semantic focus or prediction faithfulness?
- Failure Cases - Have you analyzed scenarios where reasoning supervision fails or misguides the model? Such insights could clarify the method’s limitations and robustness.

**Details Of Ethics Concerns:**

Not Applicable.

---

### Note · Authors · 2025-11-19

**Comment:**

Dear ICLR reviewers,

Thank you so much for providing insightful feedback. I am writing to formally confirm the withdrawal of my paper from the ICLR 2026 review process.

Thank you again for the valuable feedback!

**Withdrawal Confirmation:**

I have read and agree with the venue's withdrawal policy on behalf of myself and my co-authors.